# The Modulation of Fibrosis in Vocal Fold Repair: A Study on c-Met Agonistic Antibodies and Hepatocyte Growth in Animal Studies

**DOI:** 10.3390/medicina60122033

**Published:** 2024-12-10

**Authors:** Hyun-Il Shin, Hyunsu Choi, Jae-Kyun Jung, Choung-Soo Kim

**Affiliations:** 1Department of Otolaryngology-Head and Neck Surgery, THANQ Seoul Thyroid-Head and Neck Surgery Center, Seoul 06912, Republic of Korea; 2Clinical Research Institute, Daejeon St. Mary’s Hospital, Daejeon 34943, Republic of Korea; 3R&D Center for Innovative Medicines, Helixmith Co., Ltd., Seoul 07794, Republic of Korea; 4Department of Otolaryngology-Head and Neck Surgery, College of Medicine, The Catholic University of Korea, Seoul 06591, Republic of Korea

**Keywords:** vocal fold regeneration, wound healing, scar, hepatocyte growth factor, c-Met agonistic antibody

## Abstract

*Background and Objectives*: Damage to the vocal folds frequently results in fibrosis, which can degrade vocal quality due to the buildup of collagen and modifications in the extracellular matrix (ECM). Conventional treatments have shown limited success in reversing fibrotic changes. Hepatocyte growth factor (HGF) and c-Met-targeting antibodies are promising due to their potential to inhibit fibrosis and promote regeneration. This research examines the effectiveness of injections containing c-Met agonistic antibodies relative to HGF in reducing fibrosis within a rat model of vocal fold injury. *Materials and Methods*: Forty-five Sprague Dawley rats were divided into three groups, which were HGF, c-Met agonistic antibody, and the control (PBS). The right vocal folds were injured and treated with HGF or c-Met agonistic antibody injections. RNA isolation and quantitative real-time PCR were performed to assess mRNA levels of fibrosis-related markers at 1 and 2 weeks post-injury. Histopathological analysis was conducted at 3 weeks to evaluate collagen and hyaluronic acid (HA) deposition. *Results*: Both the HGF and c-Met groups demonstrated reduced type III collagen mRNA expression compared to the PBS group. The c-Met group uniquely maintained fibronectin levels closer to normal. Additionally, the c-Met group showed significantly upregulated expression of hyaluronan synthase (HAS) 1 and HAS 3 at 2 weeks post-injury, indicating enhanced HA synthesis. Histological analysis showed significantly lower collagen deposition and higher HA in the c-Met group than in PBS, confirming superior anti-fibrotic effects and ECM restoration. *Conclusions*: c-Met agonistic antibody injections outperformed HGF in reducing fibrosis, upregulating HAS expression, and promoting HA deposition in injured vocal folds, highlighting its potential as a superior therapeutic approach for preventing fibrosis and enhancing ECM quality in vocal fold injuries. Further research on functional outcomes in larger models is recommended to validate these findings.

## 1. Introduction

Vocal fold vibration is essential for voice production and the quality of the produced voice depends on the composition and viscoelasticity of vocal fold tissue, particularly the extracellular matrix (ECM) of the lamina propria layer. Injury to the vocal fold often leads to scar formation. This scar, along with the changes in the ECM could cause serious voice problems. Therefore, minimizing scar formation is essential during the healing process.

Vocal fold scarring is difficult to treat effectively with current surgical methods and available tissue engineering materials [1]. The lack of appropriate treatments is largely due to the limitation of existing surgical options. which do not adequately address ECM biomechanics and tissue properties. Additionally current injectable materials fail to replicate the complex structure of the ECM [2].

Numerous attempts at vocal fold regeneration have been made, including the use of mesenchymal stem cells (MSCs) [3], human nasal inferior turbinate-derived mesenchymal stem cells (hTMSCs) [4], and conditioned media [5]. However, stem cell therapy presents several challenges, like an immunological response, cost concerns, the risks of malignant transformation, etc. [6]. While therapies using conditioned media or exosome therapy could potentially overcome some limitations of stem cell therapy, identifying the most effective therapeutic components remains difficult. Hepatocyte growth factor (HGF) has also been researched as a treatment candidate for vocal fold healing due to its anti-fibrotic properties [7,8]. HGF has been shown to contribute to prevent scar formation in the liver, kidneys, and lungs in animal models [9,10,11]. However, a short half-time (less than five min) is the limitation of its therapeutic use [12]. The c-Met protein is a transmembrane tyrosine kinase that binds with HGF. Research has shown that c-Met localizes in endothelial cells, where the interaction between HGF and c-Met facilitates multiple biological processes [13,14].

This study investigated whether injecting a c-Met agonistic antibody, which has a relative long half-time [15], can promote the regeneration of injured vocal folds, and it compared these effects to those of HGF. We used quantitative real-time polymerase chain reaction (PCR) and histologic examination to analyze the early phase of wound healing in rat vocal fold injury models [5].

## 2. Materials and Methods

### 2.1. Animal Experiments

Approval for this study was granted by the Animal Ethics Committee of the Catholic University of Korea (permit no. CMCDJ-AP-2022-001), and all procedures followed institutional care guidelines. Forty-five Sprague Dawley rats were divided among three groups, with 12 each in the HGF and c-Met experimental groups and 12 in the PBS control group, with three animals from each group reserved for histological analysis. The animals were anesthetized with ketamine hydrochloride (100 mg/kg) and xylazine hydrochloride (10 mg/kg), positioned semi-vertically on a custom platform and visualized using a pediatric endoscope. Microscissors were used to expose the thyroarytenoid muscle on the right vocal folds, while the left vocal fold served as an untreated control. Immediately following injury, 100 ng of HGF (Millipore, Bedford, MA, USA) in 50 µL PBS or 100 ng of c-Met agonistic antibody (Helixmith, Seoul, Republic of Korea) in 50 µL PBS was injected into the injury site for experimental groups. The PBS control group received an injection of 50 µL PBS alone. All injections were administered under direct visualization using a 30-gauge needle and pediatric laryngoscope.

### 2.2. RNA Isolation and Real-Time PCR of Rat Vocal Fold Tissues

The rat vocal folds were collected at 1 and 2 weeks post-injury and homogenized using a TissueLyser II (Qiagen, Valencia, CA, USA). RNA extraction was carried out with TRIzol reagent (Invitrogen, Carlsbad, CA, USA), then ribonuclease-free deoxyribonuclease I (Qiagen) was used to minimize contamination from genomic DNA. Complementary DNA (cDNA) was synthesized using a Reverse Transcriptase Premix Kit (Elpis Biotech, Daejeon, Republic of Korea) with 1 μg of RNA. Real-time PCR was conducted on an ABI 7500 FAST system (Applied Biosystems, Foster City, CA, USA) in a 20 μL reaction volume, containing Power SYBR Green PCR Master Mix (Applied Biosystems), 500 nM forward and reverse primers, and 1 μL of cDNA. mRNA expression levels were normalized against glyceraldehyde-3-phosphate dehydrogenase (GAPDH).

Primer sets used in this study are shown in Table 1.

### 2.3. Histopathological Analysis of Rat Vocal Fold Tissue

Three larynges from each group were collected for histological examination following euthanasia, three weeks after injury. The tissues were embedded in paraffin, sectioned coronally at a thickness of 4 µm using a microtome, and stained with hematoxylin and eosin. Verhoeff–Van Gieson and Masson’s trichrome were used for the detection of collagen fiber and elastic fiber, following standard pathology protocols. Prepared slides were examined under a light microscope (Eclipse TE300; Nikon, Tokyo, Japan). Densitometric analysis was conducted to quantify extracellular matrix staining, with areas measured using ImageJ software (Version 1.53e, National Institutes of Health, Bethesda, MD, USA).

### 2.4. Statistics

Results are presented as means with standard error of the mean (SEM). GraphPad Prism 5 (GraphPad, Inc., La Jolla, CA, USA) was used for conducting statistical analyses. To compare multiple groups, a one-way ANOVA was followed by Tukey’s post hoc test. A *p*-value of less than 0.05 was used to determine statistical significance.

## 3. Results

### 3.1. Gene Expression at 1 Week Post-Injury

The expression levels of mRNAs encoding hyaluronan synthase (HAS) 1, HAS 2, and HAS 3 did not differ among the normal, PBS, HGF, and c-Met groups. The mRNA expression level of procollagen type III (COL III) was significantly downregulated in both the c-Met and HGF groups compared with the PBS group, while the gene encoding fibronectin (FN) was upregulated in the PBS and HGF groups compared with the normal group. However, in the c-Met group, the mRNA level of FN expression did not increase as much as in the PBS and HGF groups, showing no significant difference compared to the normal group (Figure 1).

### 3.2. Gene Expression at 2 Weeks Post-Injury

At two weeks post-injury, the mRNA expression levels of HAS 1 and HAS 3 were significantly upregulated in the c-Met group compared to the normal group. HAS 2 expression in the c-Met group showed an increasing trend, although it did not reach statistical significance. In the HGF group, only HAS 3 mRNA expression was significantly upregulated compared to the normal group. COL III mRNA expression was significantly upregulated in the PBS group compared to the normal group and was significantly downregulated in both the HGF and c-Met groups. FN was upregulated in the PBS, HGF, and c-Met groups compared with the normal group, but FN was downregulated in the c-Met group compared to the PBS group (Figure 2).

### 3.3. Histological Examination

Histological analysis was conducted at 3 weeks post-injury. A densitometric analysis of histological samples showed that the c-Met groups exhibited stronger anti-fibrotic effects than the PBS group. Masson’s trichrome staining indicated significantly reduced collagen deposition in the c-Met groups compared to the PBS group. Although the HGF group also showed reduced collagen deposition compared to the PBS group, the difference was not statistically significant. HA was detected by alcian blue staining. The c-Met group exhibited significantly increased HA deposition compared to the PBS group. The HGF group also showed increased HA deposition compared to the PBS group, though this difference did not reach statistical significance (Figure 3).

## 4. Discussion

Scarring after vocal fold injuries is a prevalent cause of severe voice impairment. The scarring process includes three main phases, namely the inflammatory response, cellular proliferation, and tissue remodeling [16].

Inflammatory factors are produced within 4 to 8 h post-injury, after which there is an upregulation of hyaluronan synthases I and II and procollagens I and III, followed by a substantial recruitment of cells, mainly with fibroblastic characteristics, originating from the macula flava. On days 5–7 post-injury, the fibroblasts undergo differentiation myofibroblasts, leading to a disorganization of collagen and elastin bundles and a reduction in the density of elastin and hyaluronic acid. Subsequently, type III collagen is replaced by type I collagen, and the density of fibronectin increases [17,18]. Vocal fold scarring is characterized by excessive collagen deposition. Thus, reducing collagen production is crucial to decreasing scar formation. While fibronectin is necessary for wound healing, prolonged upregulated fibronectin expression after injury causes fibrosis and impairs the viscoelastic properties of the vocal fold [19]. The key component of vocal fold viscoelastic properties is hyaluronic acid; therefore, HA synthesis is important [20,21]. HAS genes have three subtypes, with HAS2 and HAS3 exhibiting higher enzymatic activity compared to HAS1 [22].

The key focus in the wound healing process is to reduce scarring and restore the normal composition of the lamina propria, which is distinct from simply accelerating wound closure. A recent study, although conducted on skin tissue, reported that blocking potassium channels promotes wound healing [23]. However, this study did not include histological analysis, making it difficult to determine the extent of scar formation compared to normal tissue. The researchers hypothesized that fibroblast migration and activation at the wound site accelerate wound healing. However, based on previous studies, such mechanisms are known to potentially exacerbate scarring [24,25].

In this study, both the c-Met and HGF groups exhibited reduced COL III mRNA expression at 1 or 2 weeks post-injury. However, in the c-Met group, FN mRNA expression did not increase relative to the normal group at 1 week post-injury and decreased compared to the PBS group at 2 weeks post-injury. Regarding HA, no differences were observed in the RNA expression levels of HAS1, HAS2, or HAS3 at 1 week post-injury. However, by 2 weeks post-injury, a significant increase in HAS3 mRNA expression was detected. This finding aligns with previous studies, which report an initial rise in HAS1 and HAS2—especially HAS2—within three days post-injury, followed by a subsequent increase in HAS3 [26]. Furthermore, the observed significant upregulation of HAS3 mRNA expression in both the HGF and c-Met groups is consistent with earlier research suggesting that HGF negatively regulates TGF-β, which is known to suppress HAS3 expression during tissue injury. After 3 weeks, the histologic examination confirmed that mRNA expression correlated with histological change. The c-Met group had less collagen and more HA deposition, demonstrating a more pronounced anti-fibrotic effect than the PBS group with significance. However, there was no statistical difference from the HGF group. In this report, we demonstrated that the c-Met antagonistic antibody performs better than the HGF and PBS group in the anti-fibrotic effects of vocal fold injury.

Currently used injectable materials for the vocal folds primarily aim to improve glottic closure, with Radiesse (Merz North America, Inc., Raleigh, NC, USA) providing long-term effects and hyaluronic acid (HA) offering short-term effects [27,28]. These materials are typically injected either into or outside the thyroarytenoid muscle to facilitate vocal fold adduction with a space-occupying effect [29]. However, their purpose differs from regenerating the lamina propria during the wound healing process of vocal fold injuries

The restoration of normal vocal fold properties is essential to the treatment of vocal fold scars.

Many attempts have been made to reduce vocal fold fibrosis following injury.

In tissue engineering, the regeneration of tissues or organs can be achieved by the combination of scaffolds, cells, and regulators under proper conditions. Growth factor therapy is one of the therapeutic strategies in regenerative medicine. Multiple studies have highlighted HGF’s efficacy as a promising growth factor therapy for vocal fold scarring, demonstrating its capacity to regulate ECM production in fibroblasts [7,14,30,31].

In vocal fold regeneration, HGF and c-Met expression, known for their anti-fibrotic effects, decrease immediately following vocal fold injury but increase again after 14 days [32]. This period is considered critical for vocal fold scar remodeling. Supplementing the diminished HGF effect, such as through external administration, may aid in preventing scar formation in the vocal folds [30,32]. However, the effectiveness of HGF might be limited by its short half-life, necessitating repeated injection or the use of slow-release scaffolds [33,34,35].

Repeated injections may damage the vocal folds, potentially hindering the healing process [36]. Slow-release scaffolds, although designed for sustained release, do not replicate the precise biomechanical properties of the native ECM, which may interfere with vocal fold vibration until fully degraded [29,37]. Additionally, scaffold degradation may trigger inflammatory responses, posing further risks [38]. Therefore, a single, long-acting injection without a scaffold would be more suitable for effective treatment. The c-Met agonistic antibody is administered in a single injection with PBS, eliminating the need for scaffold use.

Recently, studies have explored the use of HGF-transfected stem cells via viral vectors to extend the effects of HGF [39]. However, the use of stem cells poses challenges, such as immunological responses and the risk of malignant transformation, as previously noted. While autologous or homologous stem cells may reduce these risks, they present issues related to consistency and cost. Additionally, viral transfection techniques are limited by concerns over mutations and stability [40,41]. In contrast, the c-Met agonistic antibody offers a cell-free, virus-free therapeutic approach, potentially circumventing these issues.

The serum half-life of the c-Met agonistic antibody used in this study had a half-life of approximately 3 days, with persistence in the body for up to 6 days following intravenous injection [15]. Therefore, this agonistic antibody has the potential to compensate for the short half-life of HGF.

In this report, we demonstrated that the c-Met antagonistic antibody performs better than the HGF and PBS group in anti-fibrotic effects during the remodeling of injured vocal folds. However, there were some limitations; first, while the c-Met group was more effective in anti-fibrosis than the HGF group, the difference was smaller than expected. To clarify this daily examination of the activated c-Met receptor, mRNA expression of CoLIII, FN, and HAS following injection may be beneficial. Additionally, conducting histological examinations beyond three weeks post-injury would further enhance our understanding of the findings. But conducting such experiments would require a significantly larger number of experimental animals compared to the current study. Second, the regeneration of vocal fold tissue does not necessarily guarantee functional restoration of vocal fold abilities. Therefore, to evaluate the function of the recovered vocal folds (such as the symmetrical mucosa waves during phonation between control and study group vocal folds), additional experiments involving larger animal models, such as rabbits, and tools like high-speed video cameras are required. If additional experiments are conducted considering these limitations, it is believed that the c-Met agonist antibody can be used in human research.

## 5. Conclusions

The injection of a c-Met agonistic antibody into injured vocal folds showed an anti-fibrotic effect in the early phase of wound healing, demonstrating superiority over HGF injection. These findings highlight the potential of c-Met agonistic antibodies for regenerative therapy in vocal fold injuries.

## Figures and Tables

**Figure 1 medicina-60-02033-f001:**
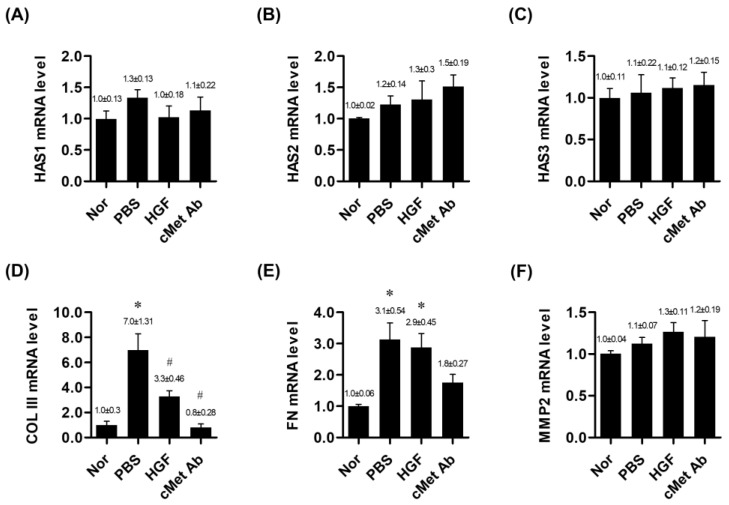
mRNA expression ratios of genes related to vocal fold tissue regeneration in the normal, PBS, and c-Met Ab groups on post-injury day 7. * *p* < 0.05 vs. Nor (normal), # *p* < 0.05 vs. PBS, *p* < 0.05 vs. HGF. (**A**–**C**) Expression levels of HAS1, HAS2, and HAS3 mRNA, respectively. (**D**) mRNA expression of COL III. (**E**) mRNA expression of FN. (**F**) mRNA expression of MMP2. Normal (Nor) tissues were compared to injured tissues treated with phosphate-buffered saline (PBS), hepatocyte growth factor (HGF), or c-Met agonistic antibody (c-Met Ab).

**Figure 2 medicina-60-02033-f002:**
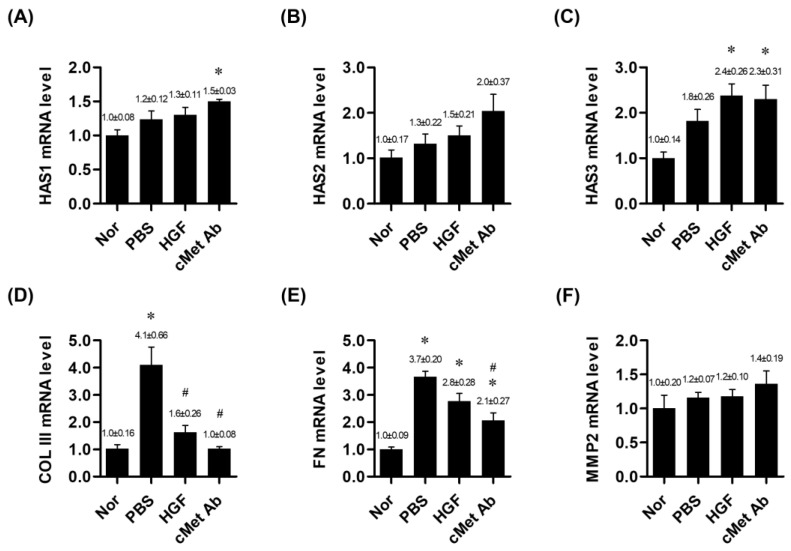
mRNA expression ratios of genes related to vocal fold tissue regeneration in the normal, PBS, and c-Met Ab groups on post-injury day 14. * *p* < 0.05 vs. Nor (normal), # *p* < 0.05 vs. PBS, *p* < 0.05 vs. HGF. (**A**–**C**) Expression levels of HAS1, HAS2, and HAS3 mRNA, respectively. (**D**) mRNA expression of COL III. (**E**) mRNA expression of FN. (**F**) mRNA expression of MMP2. Normal (Nor) tissues were compared to injured tissues treated with phosphate-buffered saline (PBS), hepatocyte growth factor (HGF), or c-Met agonistic antibody (c-Met Ab).

**Figure 3 medicina-60-02033-f003:**
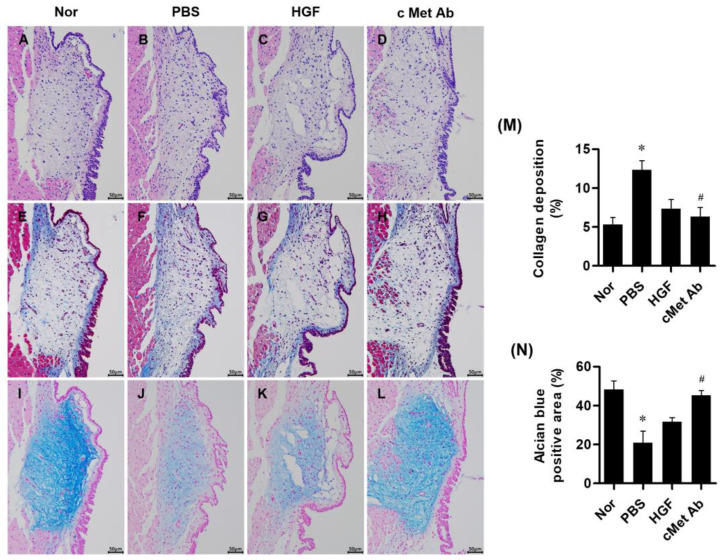
Representative images of stained vocal fold tissues at 3 weeks post-injury: hematoxylin and eosin staining (**A**–**D**), Masson’s trichrome staining for collagen (**E**–**H**), and alcian blue staining for hyaluronic acid (**I**–**L**) from the normal (Nor), PBS, HGF, and c-Met Ab groups. Scale bars = 50 µm. Quantitative analysis of collagen deposition (blue-stained collagen) (**M**) and alcian blue-positive area (blue-colored hyaluronic acids) (**N**) are shown. Original magnification: 200×; * *p* < 0.05 compared to Nor (normal), # *p* < 0.05 compared to PBS, and *p* < 0.05 compared to HGF.

**Table 1 medicina-60-02033-t001:** Primer information.

	Sequence	
Rat Genes	Forward	Reverse
*GAPDH*	5′-ACCACAGTCCATGCCATCAC-3′	5′-TCCACCACCCTGTTGCTGTA-3′
*COL III*	5′-CACTGGGGAATGGAGCAAAAC-3′	5′-ATCAGGACCACCAATGTCATAGG-3′
*FN*	5′-CGAGGTGACAGAGACCACAA-3′	5′-CTGGAGTCAAGCCAGACACA-3′
*HAS 1*	5′-TAGGTGCTGTTGGAGGAGATGTGA-3′	5′-AAGCTCGCTCCACATTGAAGGCTA-3′
*HAS 2*	5′-CCAATGCAGTTTCGGTGATG-3′	5′-ACTTGGACCGAGCCGTGTAT-3′
*HAS 3*	5′-CCTCATCGCCACAGTCATACAA-3′	5′-CCACCAGCTGCACCGTTAG-3′
*MMP2*	5′-GTCACTCCGCTGCGCTTTTCTCG-3′	5′-GACACATGGGGCACCTTCTGA-3′

Abbreviations: *GAPDH*, glyceraldehyde-3-phosphate dehydrogenase; *COL III*, collagen type III; *FN*, fibronectin; *HAS*, hyaluronan synthase; *MMP2*, Matrix metalloproteinase-2.

## Data Availability

The datasets generated during and/or analyzed during the current study are available from the corresponding author on reasonable request.

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
