# Peer review of "The Modulation of Fibrosis in Vocal Fold Repair: A Study on c-Met Agonistic Antibodies and Hepatocyte Growth in Animal Studies"

_medicina, 2024, doi:10.3390/medicina60122033_

Round 1
Reviewer 1 Report
Comments and Suggestions for Authors
Dear Authors,
I read with great interest your manuscript about vocal fold repair.
However, there are some aspects that require your attention.
The Title should clearly state that this is an animal model study.
Line 42 instead of sacring use scaring.
Figure 1, put actual values on the columns.
Figure 2, put actual values on the columns.
Figure 3, please mention the magnification level.
In the discussion section you need to expand on the use of other substances to heal leasions and to underline the superiority of your approach in promoting healing with diminished scaring. Such research was undertaken also in similar animal models, but at the level of the skin. There is a focus on healing in spite of possible scaring. One possible reference could be the work by Grigore, A.; Vatasescu-Balcan, A.; Stoleru, S.; Zugravu, A.; Poenaru, E.; Engi, M.; Coman, O.A.; Fulga, I. Experimental Research on the Influence of Ion Channels on the Healing of Skin Wounds in Rats. Processes 2024, 12, 109. https://doi.org/10.3390/pr12010109
Take into consideration Radiese procedures that are currently undertaken at the level of the vocal folds and try to compare your results with the direct injection of HA in the vocal folds.
Before the conclusions please insert a small paragraph about the limitations of this study and the need to further translate the findings from animal models to human subjects.
There are many abbreviations in the text, a list of abbreviations at the end of the manuscript could help increase the readability of the manuscript.
Please update some of the references, there is a title from 1999. I am sure that there are newer articles in the same subject that are worth mentioning.
Looking forward to receiving the improved version of the manuscript.
Author Response
I read with great interest your manuscript about vocal fold repair.
However, there are some aspects that require your attention.
1.The Title should clearly state that this is an animal model study.
Response:”Modulation of Fibrosis in Vocal Fold Repair: A Study on c-MET Agonistic Antibodies and Hepatocyte Growth in animal studies”
2.Line 42 instead of sacring use scaring.
Response : Thank you for the comment we have corrected the spelling of “sacring “ to “scaring”
3.Figure 1, put actual values on the columns.
Response : We have inserted actual values on the columns
4.Figure 2, put actual values on the columns.
Response : We have inserted actual values on the columns
5.Figure 3, please mention the magnification level.
Response : We have inserted magnification level in the figure 3 legend
6.In the discussion section you need to expand on the use of other substances to heal leasions and to underline the superiority of your approach in promoting healing with diminished scaring. Such research was undertaken also in similar animal models, but at the level of the skin. There is a focus on healing in spite of possible scaring. One possible reference could be the work by Grigore, A.; Vatasescu-Balcan, A.; Stoleru, S.; Zugravu, A.; Poenaru, E.; Engi, M.; Coman, O.A.; Fulga, I. Experimental Research on the Influence of Ion Channels on the Healing of Skin Wounds in Rats. Processes2024, 12, 109. https://doi.org/10.3390/pr12010109
Response : Thanks for your comments
In the introduction and discussion sections, we addressed the limitations of existing stem cell therapies, as well as those of conditioned media and exosome therapies derived from stem cells. In the discussion, we focused on growth factor therapy as one of the treatment modalities for vocal fold regeneration. We highlighted the advantages of the c-MET agonistic antibody in comparison to previous studies utilizing HGF.
Additionally, we elaborated on the distinction between accelerating wound healing and reducing scarring by incorporating insights from a recent paper mentioned by the reviewer, as follows
“The key focus in the wound healing process is to reduce scarring and restore the normal composition of the lamina propria, which is distinct from simply accelerating wound closure. A recent study, although conducted on skin tissue, reported that blocking potassium channels promotes wound healing. However, this study did not include histological analysis, making it difficult to determine the extent of scar formation compared to normal tissue. The researchers hypothesized that fibroblast migration and activation at the wound site accelerate wound healing. However, based on previous studies, such mechanisms are known to potentially exacerbate scarring”
7.Take into consideration Radiese procedures that are currently undertaken at the level of the vocal folds and try to compare your results with the direct injection of HA in the vocal folds.
Response : Thanks for your comments, we agreed your opinion that inserting current injection material, and describing the difference with our studies
We inserted “Currently used injectable materials for the vocal folds primarily aim to improve glottic closure, with Radiesse (Merz North America, Inc., USA) providing long-term effects and hyaluronic acid (HA) offering short-term effects. These materials are typically injected either into or outside the thyroarytenoid muscle to facilitate vocal fold adduction. However, their purpose differs from regenerating the lamina propria during the wound healing process of vocal fold injuries.”
8.Before the conclusions please insert a small paragraph about the limitations of this study and the need to further translate the findings from animal models to human subjects.
Response : We have already described about the limitations of this study as follows.
“However, there were some limitations, first while the c-Met group was more effective in anti-fibrosis than the HGF group, the difference was smaller than expected. To clarify this daily examination of the activated c-Met receptor, , mRNA expression of CoLIII ,FN and HAS following injection may be beneficial. Additionally, conducting histological examinations beyond three weeks post-injury would further enhance our understanding of the findings. But conducting such experiments would require a significantly larger number of experimental animals compared to the current study. Second Regeneration of vocal fold tissue does not necessarily guarantee functional restoration of vocal fold abilities. Therefore, to evaluate the function of the recovered vocal folds(such as the symmetrical mucosa waves during phonation between control and study group vocal folds), additional experiments involving larger animal models, such as rabbits, and tools like high-speed video cameras are required”
And added following sentence
“If additional experiments are conducted on these limitations, it is believed that c-met agonist antibody can be used in human research.”
9.There are many abbreviations in the text, a list of abbreviations at the end of the manuscript could help increase the readability of the manuscript.
Response: we have added a list of abbreviations in appendix
|
Abbreviations |
Definition |
|
ECM |
Extracellular matrix |
|
HGF |
Hepatocyte growth factor |
|
PBS |
Phosphate buffered saline |
|
RNA |
Ribonucleic Acid |
|
HA |
Hyaluronic acid |
|
HAS |
Hyaluronan synthase |
|
MSC |
Mesenchymal stem cells |
|
PCR |
polymerase chain reaction |
|
DNA |
Deoxyribonucleic acid |
|
COLIII |
Procollagen type III |
|
FN |
Fibronectin |
|
TGF-β |
Transforming growth factors-beta superfamily |
10 . Please update some of the references, there is a title from 1999. I am sure that there are newer articles in the same subject that are worth mentioning.
Response : Thanks for your opinion . we update the reference” Siiskonen H, Oikari S, Pasonen-Seppänen S, Rilla K. Hyaluronan synthase 1: a mysterious enzyme with unexpected functions. Front Immunol. 2015;6:43.”
Thanks you for taking your valuable time to review this manuscript
Reviewer 2 Report
Comments and Suggestions for Authors
Introduction
1. The composition and viscoelasticity of the ECM in the vocal fold are critical. Injury can result in the replacement of normal tissue with fibrotic scar, leading to a degradation of voice quality. cite doi:10.3791/61327.
2. Current medical treatments for vocal fold scarring are limited by the lack of surgical techniques and injectable materials suitable for restoring the lost, complex structure of the ECM. cite doi:10.1159/000456684.
3. Give an elaborated reason behind the study of c-Met agonistic antibodies as a potential therapeutic approach, including advantages over the use of HGF such as longer half-life and thereby overcoming limitations associated with HGF.
Materials and Methods
1. Give the rationale for selecting doses of HGF (100 ng) and c-Met agonistic antibody (100 ng) in this study.
2. More explanation on histological evaluation is needed, such as the criteria to assess collagen deposition and HA deposition.
Results:
1. The possible causes for these discrepancies in mRNA expression profile in c-Met and HGF groups are discussed, especially in maintaining the level of fibronectin closer to normal in the c-Met group.
2. Discuss possible reasons for the poorer anti-fibrotic differential between the c-Met and HGF groups than might have been anticipated.
Discussion:
1. The use of c-Met agonistic antibody may be envisioned to confer some advantages over therapies based on the use of stem cells, by avoiding associated immunological responses and the risk of malignant transformation and discuss all potential biomarkers.
2. Discuss the relevance of observed HAS3 mRNA expression upregulation within the c-Met group and how this could explain the better restoration of the ECM.
3. Do mention the limitations regarding functional assessments and give more specific recommendations for further studies to be conducted with larger animal models regarding functional outcomes after the treatment with a c-Met agonistic antibody.
Author Response
introduction
1. The composition and viscoelasticity of the ECM in the vocal fold are critical. Injury can result in the replacement of normal tissue with fibrotic scar, leading to a degradation of voice quality. cite doi:10.3791/61327.
Response : The manuscript of cite doi:10.3791/61327. “Preparation of the Rat Vocal Fold for Neuromuscular Analyses” described high variability in the depth of laryngeal landmark.
We agree with the points raised in this paper. However, our experiment analyzed the vocal fold muscle and lamina propria in the coronal sections where these structures are most clearly visible, despite variations in the rat larynx. While landmark depth may vary significantly between individual specimens, it has been reported that the composition of the lamina propria in the vocal folds varies depending on the age of the rat(REF) Therefore, in this study, we used rats of the same age and size to minimize inter-individual differences, allowing for more consistent comparisons of histological findings across the groups.
(REF) Ann Otol Rhinol Laryngol Author manuscript; available in PMC: 2009 Nov 25.
Published in final edited form as: Ann Otol Rhinol Laryngol. 2009 Oct;118(10):735–741. doi: 10.1177/000348940911801009
“Age-Associated Changes in the Expression and Deposition of Vocal Fold Collagen and Hyaluronan”
2. Current medical treatments for vocal fold scarring are limited by the lack of surgical techniques and injectable materials suitable for restoring the lost, complex structure of the ECM. cite doi:10.1159/000456684.
Reponse : The manuscript of cite doi:10.1159/000456684. “Injection Laryngoplasty for Management of Neurological Vocal Fold Immobility “ described about injection laryngoplasty for vocal cord movement disorders
Currently, vocal injection materials are commercially used primarily for treating vocal fold paralysis or improving glottic gap. However, materials designed to regenerate the extracellular matrix (ECM) and reduce vocal fold scarring following injury are still in the experimental stages. This study represents one such experimental approach. These points have been included in the discussion section. As follows
“Currently used injectable materials for the vocal folds primarily aim to improve glottic closure, with Radiesse( ) providing long-term effects and hyaluronic acid (HA) offering short-term effects. These materials are typically injected either into or outside the thyroarytenoid muscle to facilitate vocal fold adduction with space occupying effect. However, their purpose differs from regenerating the lamina propria during the wound healing process of vocal fold injuries.”
3.Give an elaborated reason behind the study of c-Met agonistic antibodies as a potential therapeutic approach, including advantages over the use of HGF such as longer half-life and thereby overcoming limitations associated with HGF.
Response : Thanks for your opinion. Generally therapeutic antibodies have long half-lives in the bloodstream, lack toxicity, and are exceptionally potent and specific for their molecular targets. In this study we have describe introduction and discussion
Materials and Methods
1. Give the rationale for selecting doses of HGF (100 ng) and c-Met agonistic antibody (100 ng) in this study.
Response : We agree with the reviewer’s comment. Before initiating the experiment using the c-MET agonistic antibody developed by Helixmith.co.kr. we inquired with the manufacturer. They explained that the c-MET antibody was designed to achieve the same effects as HGF at an equivalent concentration.
2.More explanation on histological evaluation is needed, such as the criteria to assess collagen deposition and HA deposition.
Response : Thank you for the comment. Unfortunately There are no absolute criteria for collagen and HA deposition within tissues. Furthermore, collagen and HA levels in the vocal folds vary between individuals and change with age. To minimize these differences, we used rats of the same age and size. For histological analysis, the uninjured left vocal fold was designated as the normal control group, while the PBS group served as the injury control. The HGF and c-MET groups were then compared against these controls
Results
1. The possible causes for these discrepancies in mRNA expression profile in c-Met and HGF groups are discussed, especially in maintaining the level of fibronectin closer to normal in the c-Met group.
Response: Both COL III and FN are essential in the wound healing process; however, excessive production of these components can increase tissue stiffness over time. In the vocal folds, increased stiffness of the lamina propria ultimately degrades vocal quality. In this study, the groups treated with c-MET and HGF showed a statistically significant reduction in COL III expression compared to the PBS group in mRNA expression analysis at both one and two weeks. Although FN expression did not show statistical significance, it demonstrated a decreasing trend. Notably, FN expression in the c-MET group was observed to be lower than in the HGF group, likely due to the longer duration of action of the c-MET antibody compared to HGF. This observation has been discussed in detail in the discussion section.
2.Discuss possible reasons for the poorer anti-fibrotic differential between the c-Met and HGF groups than might have been anticipated.
Response : The anti-fibrotic effect of c-MET, as assessed by mRNA expression levels, was found to be greater than that of HGF, and this was confirmed through histological examination. However, since mRNA levels were only evaluated at one and two weeks, continuously monitoring the mRNA expression levels of COL III, FN, and HAS over a two-week period could provide a more comprehensive understanding. Additionally, conducting histological examinations beyond three weeks post-injury would further enhance our understanding of the findings. This limitation has been addressed in the discussion section
Discussion:
1. The use of c-Met agonistic antibody may be envisioned to confer some advantages over therapies based on the use of stem cells, by avoiding associated immunological responses and the risk of malignant transformation and discuss all potential biomarkers.
Response : In the introduction, we addressed the limitations of existing stem cell therapies, as well as those of conditioned media and exosome therapies derived from stem cells. Additionally, in the discussion section, we further elaborated as follows:
“However, the use of stem cells poses challenges, such as immunological responses and the risk of malignant transformation, as previously noted. While autologous or homologous stem cells may reduce these risks, they present issues related to consistency and cost.”
In the discussion, we focused on growth factor therapy as one of the various therapeutic approaches for vocal fold regeneration, highlighting the advantages of c-MET agonistic antibodies in comparison to previous studies utilizing HGF.
For cases where malignant transformation occurs after stem cell therapy, potential biomarkers can provide evidence. Genetically, if the stem cells were derived externally (transplanted cells), analyzing the HLA (Human Leukocyte Antigen) pattern of cancer cells and comparing it with the patient’s unique HLA can indicate that the origin of the cancer is the transplanted stem cells if identical HLA patterns are detected. Additionally, if the stem cells used in therapy included genetic tags or vectors, the presence of the same tags or viruses in the cancer cells could confirm that the transplanted stem cells were the source of the malignancy.
2. Discuss the relevance of observed HAS3 mRNA expression upregulation within the c-Met group and how this could explain the better restoration of the ECM.
Response : The mRNA expression patterns of HAS1, HAS2, and HAS3 have been detailed in the discussion section.
“Regarding HA no differences were observed in the RNA expression levels of HAS1, HAS2, or HAS3 at 1 week post-injury. However, by 2 weeks post-injury, a significant increase in HAS3 mRNA expression was detected. This finding aligns with previous studies, which report an initial rise in HAS1 and HAS2—especially HAS2—within three days post-injury, followed by a subsequent increase in HAS3.[23] Furthermore, the observed significant upregulation of HAS3 mRNA expression in both the HGF and c-Met groups is consistent with earlier research suggesting that HGF negatively regulates TGF-β, which is known to suppress HAS3 expression during tissue injury.”
Daily mRNA analysis following the injury might have demonstrated an earlier increase in HAS1 and HAS2 expression, potentially within the first three days. However, in this study, mRNA analysis was conducted only at one and two weeks post-injury, and this limitation has been addressed in the discussion section
3.Do mention the limitations regarding functional assessments and give more specific recommendations for further studies to be conducted with larger animal models regarding functional outcomes after the treatment with a c-Met agonistic antibody.
Response : Thanks for your opinion we have added the limitation of this studies
“However, there were some limitations, first while the c-Met group was more effective in anti-fibrosis than the HGF group, the difference was smaller than expected. To clarify this daily examination of the activated c-Met receptor, mRNA expression of CoLIII ,FN and HAS following injection may be beneficial. Additionally, conducting histological examinations beyond three weeks post-injury would further enhance our understanding of the findings. But conducting such experiments would require a significantly larger number of experimental animals compared to the current study.
Second Regeneration of vocal fold tissue does not necessarily guarantee functional restoration of vocal fold abilities. Therefore, to evaluate the function of the recovered vocal folds (such as the symmetrical mucosa waves during phonation between control and study group vocal folds), additional experiments involving larger animal models, such as rabbits, and tools like high-speed video cameras are required. If additional experiments are conducted on these limitations, it is believed that c-met agonist antibody can be used in human research
Thanks you for taking your valuable time to review this manuscript